# Effects of Long-Term Blue Light Exposure on Body Fat Synthesis and Body Weight Gain in Mice and the Inhibitory Effect of Tranexamic Acid

**DOI:** 10.3390/ijms26125554

**Published:** 2025-06-10

**Authors:** Keiichi Hiramoto, Hirotaka Oikawa

**Affiliations:** Department of Pharmaceutical Sciences, Suzuka University of Medical Science, 3500-3 Minamitamagaki, Suzuka 513-8607, Mie, Japan; oikawah@suzuka-u.ac.jp

**Keywords:** blue light, Bmal 1, mTORC 1, SREBP 1, Sirt1, weight changes, tranexamic acid

## Abstract

Humans are continuously exposed to blue light from sunlight, computers, and smartphones. While blue light has been reported to affect living organisms, its role in fat synthesis and weight changes remains unclear. In this study, we investigated the effects of prolonged blue light exposure on weight changes in mice and the protective role of tranexamic acid (TA). Mice were exposed daily to blue light from a light-emitting diode for five months. Blue light exposure led to increased fat mass and body weight. The expression of the clock genes arnt-like 1 (Bmal1) and Clock was reduced in the brain and muscle of exposed mice. In addition, reduced Sirt1 and increased mammalian target of rapamycin complex 1 (mTORC1)/sterol regulatory element-binding protein 1 (SREBP1) were observed. The levels of liver X receptor a and liver kinase B1/5′AMP-activated protein kinase a1, both involved in SREBP1-mediated lipogenesis, were also elevated. TA treatment prevented the blue light-induced suppression of Bmal1/Clock and modulated the subsequent series of signal transduction. These findings suggest that prolonged blue light exposure suppresses the clock gene Bmal1/Clock, reduces Sirt1, and activates lipogenic pathways, contributing to weight gain. TA appears to regulate clock gene expression and mitigate blue light-induced weight gain.

## 1. Introduction

Exposure to light at night has been suggested to impact health negatively. Animal studies show that nighttime light exposure disrupts circadian rhythms, reduces sleep quality, increases food intake, and promotes weight gain [1]. In humans, women exposed to artificial light from electronic devices at night were more likely to gain weight than those who were not [2]. In a study by Park YMM et al., women who slept with a TV or room lights on were more likely to gain approximately 5 kg compared with those unexposed to artificial light at night. This association persisted after adjusting for factors such as residence, income, caffeine and alcohol intake, depression, and stress. However, a direct causal relationship between nighttime light exposure and weight gain remains unconfirmed [2].

In modern times, the human body is frequently exposed to blue light (visible light: 380–495 nm) emitted from light sources such as light-emitting diodes (LEDs), fluorescent lamps, incandescent light, and digital screens [3,4]. Exposure to blue light disrupts circadian rhythms through non-visual physiological effects and causes retinal damage [5]. Furthermore, exposure to blue light at night suppresses melatonin secretion, increases core temperature and heart rate, and reduces sleepiness [6,7,8]. We previously reported that blue light affects the ciliary muscle and induces eye strain [9]. Blue light also affects the skin. Exposure increases reactive oxygen species and collagen-degrading enzymes (matrix metalloproteinase), contributing to skin aging [10,11]. Additionally, increased melanin pigmentation and changes in skin color have been reported [12,13]. Blue light exposure has been implicated in various forms of biological damage; however, its effects on weight gain remain poorly understood.

In addition, tranexamic acid (trans-4-aminomethylcyclohexanecarboxylic acid) is a medical amino acid with antiplasmin activity, which exerts hemostatic, anti-inflammatory, and anti-allergic effects. The action of tranexamic acid depends on the binding of plasmin and plasminogen to fibrin. As a result, tranexamic acid exhibits fibrinolysis inhibition, arachidonic acid dissociation inhibition, prostaglandin and leukotriene production inhibition, active oxygen dissociation inhibition in neutrophils, and histamine dissociation inhibition in mast cells [14,15,16]. In addition, tranexamic acid has the effect of reducing age spots, melasma, and pigmentation caused by ultraviolet (UV) rays [17,18].

In this study, we investigated the effects of blue light exposure on weight gain in mice and evaluated the protective role of tranexamic acid (TA), a compound reported to mitigate blue light-induced damage in living organisms [9,19,20].

We further checked whether there were sex differences in weight gain. As a result, we found that there was no difference between males and females in the effect of blue light exposure on body weight (Appendix A). Therefore, in this study, we decided to use males, considering the effect of the sexual cycle in females.

## 2. Results

### 2.1. Effects of TA Treatment on Body Weight, Food Intake, Fat Accumulation, and Lipid Profiles in Blue Light-Exposed Mice

TA (12 mg/kg) was orally administered to mice exposed to blue light thrice a week for five months. By the end of the experiment, blue light-exposed mice showed increased body weight and food intake compared with the control group. In contrast, TA-treated mice showed no significant difference in body weight or food intake compared with the control group (Figure 1a,b). Furthermore, blue light exposure increased total body fat, subcutaneous fat, visceral fat, and triglyceride levels, all of which were reduced by TA treatment (Figure 1c–f). Cholesterol levels showed a non-significant increase with blue light exposure but were reduced by TA treatment (Figure 1g).

### 2.2. Effects of TA Treatment on p53, Clock Genes (Brain and Muscle Arnt-like 1 (Bmal1) and Clock), Sirt,1 and Sterol Regulatory Element-Binding Protein 1 (SREBP1) in Blue Light-Exposed Mice

We investigated clock genes that are affected by blue light exposure, as well as related regulatory genes. The clock genes Bmal1 and Clock were decreased by blue light exposure and increased by TA treatment (Figure 2b,c). Similarly, Sirt1 was decreased by blue light exposure and increased by TA administration (Figure 2d). In contrast, p53 (Figure 2a) and SREBP1 (Figure 2e) was increased by blue light exposure and decreased by TA administration.

### 2.3. Effects of TA Treatment on Nicotinamide Mononucleotide Adnylyltransferase (NMNAT) Activity, Nicotimamide Phosphoribosyltransferase (NAMPT) Activity, and Nicotinamide Asenine Dinucleotide^+^ (NAD^+^) in Blue Light-Exposed Mice

To investigate the relationship between clock genes and Sirt1, we examined NMNAT activity, NAMPT activity, and NAD^+^. These genes were decreased by blue light exposure and increased by TA administration compared with the control group (Figure 3a–c).

### 2.4. Effect of TA Treatment on Liver X Receptor α (LXRα), Liver Kinase B1 (LKB1), 5′AMP-Activated Protein Kinase a1 (AMPKa1), and Mammalian Target of Rapamycin Complex 1 (mTORC1) in Blue Light Irradiation Mice

mTORC1 was measured as a factor that increases AREBP1. Furthermore, we investigated LXRα, LKB1, and AMPKa1, which are involved in the activation of mTORC1. mTORC1 was estimated from the merged region of mTOR and Raptor co-staining (Figure 4a). mTORC1 and LXRα were increased by blue light exposure compared with the control group (Figure 4a,b). In contrast, it was decreased by TA administration. The expression of LKB1 (Figure 4c) and AMPK (Figure 4d) was decreased by exposure to blue light and restored by the administration of TA.

## 3. Discussion

In this study, chronic exposure to blue light for five months increased food intake, fat percentage, and body weight in mice. Blue light exposure reduces the expression of the clock genes Bmal1 and Clock. Bmal1/Clock suppresses the expression of NAMPT, thereby inhibiting the metabolism of NAM to NMN. In addition, the production of NAD is reduced, thereby inhibiting the production of Sirt1. The reduction in Sirt1 activates LXRa and simultaneously reduces LKB1. The reduction in LKB1 increases mTORC1/AREBP1 and increases lipid synthesis. The increased lipid synthesis is thought to have caused the weight gain of the mice. TA is proposed to suppress the increase in p53, which is mostly located upstream of blue light exposure, and increases Bmal1/Clock, thereby suppressing lipid synthesis.

Blue light irradiation induces a decrease in the expression of Bmal1 and CLOCK and an increase in Cry1 expression [21]. In this study, we also observed a decrease in the expression of Bmal1 and CLOCK in blue light-exposed mice. The NAD^+^ constitutive system and clock genes are closely related, and a decrease in the Bmal1/CLOCK complex inhibits the NAD^+^ constitutive system. Previous reports suggest that long-term UVA exposure increases p53 protein levels and decreases clock gene expression [22]. p53 inhibits the binding of Per2 to the Bmal1/Clock complex, inhibiting the transcription of clock genes, including Pers, Crys, and rev-Erbs, as well as NAMPT [23,24]. NAMPT synthesizes NAD^+^ and increases Sirt1 levels [24,25]. The up-regulation of p53 production inhibits Per and Bmal1/Clock complexes, reduces NAMPT levels, and consequently lowers NAD^+^ levels. In this study, the p53 protein levels in the liver were increased by blue light exposure, suggesting that a mechanism similar to UVA irradiation was at play. These findings indicate that blue light may elevate p53, disrupt the Bmal1/clock complex, and suppress NAMPT expression, ultimately decreasing intracellular NAD^+^ [24].

NAD^+^ plays a central role in age-related and metabolic diseases such as cancer, diabetes, and obesity [25,26,27,28]. NAD^+^ synthesis involves precursors such as nicotinamide (NAM), nicotinic acid, and nicotinamide riboside, with NAMPT and NMNAT playing key roles in its production. NAMPT converts NAM to nicotinamide mononucleotide, which is continuously converted to NAD^+^ by NMNAT [25,26,27,28]. Sirt1 is involved in survival and acts as a catalyst promoted by NAD^+^ [29]. Sirt1 is involved in the regulation of the cell cycle, cell senescence, apoptosis, stress resistance, and metabolism and supports the physiological responses seen in calorie restriction. The overexpression of Sirt1 in mice showed improved glucose, cholesterol, and lipid metabolism, playing an important role in metabolic regulation or normalization. In lipid metabolism, Sirt1 promotes homeostasis through the deacetylation of LXR, a regulator of cholesterol and lipid synthesis [30]. In this study, LXRa expression was increased in blue light-exposed mice, potentially contributing to enhanced hepatic lipid and cholesterol synthesis. This series of signaling may lead to an increase in body weight due to blue light. Furthermore, increased intracellular NAD+ production activates the NAD^+^-dependent acetyltransferase SIRT1, which activates LKB1, an enzyme that phosphorylates and activates AMPK, by deacetylating it, leading to the activation of AMPK [31]. AMPK suppresses mTORC1 activity by activating TSC1/2 and decreasing the activity of Rheb (an mTORC1 activator) [32]. Consistent with this pathway, blue light exposure reduced AMPK activity by decreasing Sirt1 levels, which in turn increased the expression of mTORC1. mTORC1 promotes fat synthesis in the liver through the activation of SREBP1, a transcription factor that plays a central role in regulating lipid metabolism, such as cholesterol and triglycerides [33]. These findings suggest that blue light may promote fat accumulation via the mTORC1/SREBP1 axis.

In this study, TA, a drug that acts on blood vessels, was shown to reduce weight gain caused by blue light exposure. Blue light exposure significantly reduced the expression of Bmal1/clock. Bmal1 and sympathetic nerves are inversely correlated, and when Bmal1 levels decrease, norepinephrine levels increase [34]. Blue light exposure activates the sympathetic nervous system and induces the secretion of large amounts of noradrenaline [35]. Noradrenaline induces the expression of CCL2 and ICAM1 in skin vascular endothelial cells via β2-adrenergic receptors, promoting the migration of neutrophils from blood vessels to tissues and increasing the number of neutrophils in tissues [36,37]. Neutrophils generate ROS through NADPH oxidase and myeloperoxidase [38,39,40]. Under blue light irradiation, an increase in neutrophils promotes an increase in ROS. In contrast, p53 expression is related to intracellular ROS, and p53 expression also increases with ROS [41]. We speculate that blue light exposure elevated ROS levels by increasing neutrophil activity, which could account for the observed rise in p53. Given that *p53* is sensitive to intracellular oxidative stress, this mechanism may contribute to downstream effects on circadian gene expression. TA has the effect of suppressing the increase in neutrophils caused by blue light irradiation [19,42]. Therefore, TA may improve downstream signals of neutrophils/ROS/p53.

Furthermore, blue light exposure is closely related to sleep. Exposure to artificial light at night shortens sleep time. This indirectly affects the risk of obesity and disrupts food-related hormones, leading to increased food intake [2]. In addition, reduced sleep time leads to reduced physical activity [43]. Sleep helps balance the hormones leptin and ghrelin, which control appetite [44]. Sleep deprivation reduces leptin (which suppresses appetite) and increases ghrelin (which stimulates appetite), resulting in increased appetite and overeating. A lack of sleep also triggers a stress response that increases cortisol levels. Cortisol causes the body to conserve energy and promotes fat storage [45]. In this study, we did not analyze the relationship between blue light exposure and sleep. Because mice are nocturnal animals, our blue light protocol (administered from 10:00 a.m. to 10:10 p.m.) may have interfered with their sleep. While no significant changes in ghrelin and leptin were observed, corticosterone levels were elevated under blue light exposure, indicating the activation of a stress response (Appendix A). Therefore, corticosterone may also contribute to increased fat synthesis and weight gain. Future studies should examine sleep patterns in this model, as well as the potential effects of TA on sleep regulation.

## 4. Materials and Methods

### 4.1. Animal Experiments

Six-week-old male specific pathogen-free (SPF) Institute of Cancer Research mice (ICR: SLC, Hamamatsu, Shizuoka, Japan) were used as experimental animals. The mice were housed individually at 23 ± 1 °C under SPF and stress-free conditions with a 12 h light/12 h dark cycle. The mice were divided into control (white light irradiation), blue light irradiation, and blue light irradiation + TA administration groups, each group consisting of 5 mice. The light sources used were LED blue light (wavelength: 380–500 nm, peak emission: 479 nm, 20 kJ/m^2^, ISLM-150X150-BB, CCS, Kyoto, Japan), and LED white light (6 kJ/m^2^, ISLM-150X150-HWW, CCS, Kyoto, Japan). The energy content of each LED light was measured using a photoanalyzer (LA-105, Nippon Medical Instruments, Osaka, Japan). The control group was irradiated with LED white light. Mice were exposed daily to the designated light source for 10 min over 5 months. The irradiation energy of blue light was selected based on the results of a preliminary test where the energy amount was varied between 10, 20, and 40 kJ/m^2^, and the greatest increase in body weight was observed when irradiated with 20 kJ/m^2^ (Appendix A). This amount of energy reflects typical daily environmental exposure. Furthermore, no change was observed in the control (white light) + TA administration group compared with the control group; therefore, only three groups were included in the study: control, blue light irradiation group, and blue light + TA administration group. This study was approved by the Suzuka University of Medical Science Animal Experiment Ethics Committee on 25 September 2019, and was conducted in strict accordance with the Suzuka University of Medical Science Guide for the Care and Use of Laboratory Animals (approval number: 34). All surgeries were performed on mice under pentobarbital anesthesia. Efforts were made to minimize animal suffering. To examine whether blue light exposure and TA administration would cause liver damage in mice, we measured plasma aspartate transaminase (AST) and alanine transaminase (ALT) levels, which are markers of liver damage. As a result, blue light exposure and TA administration had no effect on AST and ALT levels, and no liver damage was induced (Appendix A).

### 4.2. Tranexamic Acid Treatment

Approximately 12 mg kg^−1^ of TA (Daiichi Sankyo Healthcare Co., Ltd., Tokyo, Japan) in distilled water was administered orally three times a week for 5 months. Control and blue light-exposed mice were administered distilled water [46]. This dose was converted from the human therapeutic dose and showed no adverse effects or changes in body weight, histology, or skin biochemistry in preliminary testing, suggesting that TA is potentially nontoxic to mice at the dose administered.

### 4.3. Measurement of Body Fat Percentage, Subcutaneous Fat Percentage, and Visceral Fat Percentage

Mice were anesthetized via an intraperitoneal injection of ketamine hydrochloride (100 mg/kg) and xylazine hydrochloride (10 mg/kg). The anesthetized mice underwent whole-body tomography imaging using a 3D micro X-ray CT system (Rigaku, Tokyo, Japan) under the following conditions: tube voltage, 90 kV; tube current, 150 μA; imaging time, 17 s; field of view/image size, φ9.6 × H9.6 mm; and resolution/voxel size, 20 × 20 × 20 μm^3^. Tomographic data were processed into CT and DICOM data using the i-VIEW-R micro X-ray CT software ver 1.73 (J. MORITA MFG.CORP, Kyoto, Japan), and body fat percentage was analyzed based on the data using the CTAtlas body fat analysis software ver. 2.03 (Rigaku, Tokyo, Japan). The region of interest for fat measurement was defined as the abdominal area, spanning from the lower thoracic vertebrae through the lumbar vertebrae to the upper sacral vertebrae. Fat volume was quantified and expressed as the percentage of body fat, subcutaneous fat, and visceral fat within the measurement volume. Subcutaneous fat and visceral fat were visually differentiated in the analysis, with subcutaneous fat displayed in orange and visceral fat in yellow.

### 4.4. Preparation and Staining of the Liver

On the last day of the experiment, liver samples were harvested under anesthesia. Liver samples were fixed in 4% phosphate-buffered paraformaldehyde, embedded in Tissue Tek, OCT compound (Sakura Finetek, Tokyo, Japan), and cryosectioned. These sections were stained using antibodies for immunohistological analysis as previously described [47]. Liver specimens were incubated with primary antibodies, either mouse monoclonal anti-LXRa (1:100, Abcam, Cambridge, MA, USA), rabbit polyclonal anti-LKB1 (1:100, Proteintech Group, Rosemont, IL, USA), mouse monoclonal anti-AMPKa1 (1:100, Proteintech Group), mouse monoclonal anti-mTOR (1:100, Proteintech Group, Rosemont, IL, USA), or rabbit polyclonal anti-Raptor (1:100, Proteintech Group). Sections were incubated with fluorescein isothiocyanate-conjugated anti-rabbit or anti-mouse secondary antibodies (1:30, Daco Cytomation, Glostrup, Denmark). The fluorescence intensity of the merged mTOR and Raptor was calculated from four random fields of view of a given area using ImageJ software version 1.53 (National Institutes of Health, Bethesda, MD, USA). The original files were converted to monochrome 8-bit files. A luminosity threshold was then set independently. Areas above the threshold were measured for each sample. These areas were defined as “intensity” in this study.

### 4.5. Western Blotting Analysis of the Liver

Liver samples were homogenized in lysis buffer (Kurabo, Osaka, Japan) and then centrifuged to obtain the supernatant. Western blotting was performed as previously described [48]. Membranes were incubated with primary antibodies against Bmal1 (1:1000; Cell Signaling Technology, Danvers, MA, USA), Clock (1:500; Medical & Biological Laboratories, Nagoya, Aichi, Japan), Sirt1 (1:1000; Cell Signaling Technology), SREBP1 (1:1000; Proteintech Group), and β-actin (1:5000; Sigma-Aldrich, St. Louis, MO, USA) for 1 h at room temperature. β-actin was added as a loading control. Plus2 Pre-Stained Protein Standard (Life Technologies, Carlsbad, CA, USA) was also applied to the membrane as a protein marker. After primary antibody incubation, the membranes were washed and incubated with horseradish peroxidase-conjugated secondary antibodies (Novex, Frederick, MD, USA). Immune complexes were detected using ImmunoStar Zeta reagent (Wako Pure Chemical Industries, Osaka, Japan), and images were acquired using Multi Gauge software ver. 3.0 (Fujifilm, Greenwood, SC, USA).

### 4.6. Measurement of Triglyceride, Cholesterol, NAMPT Activity, NMNAT Activity, NAD^+^, and p53 Levels in Mouse Liver

Liver samples were collected on the final day of the experiment. The liver was isolated and homogenized in a lysis buffer (Kurabo, Osaka, Japan). The tissue extracts were centrifuged (Tomy MX-201) at 10,000 rpm. The supernatants were collected for the assay. Commercial enzyme-linked immunosorbent assay kits were used, according to the manufacturers’ instructions, to measure the following: triglyceride and cholesterol (FUJIFILM Wako Shibayagi, Shibukawa, Gunma, Japan), NAMPT activity and NMNAT activity (Medical & Biological Laboratories), NAD^+^ (Cel Biolabs, Inc., Tokyo, Japan), and p53 (Abcam). Optical density was measured using a microplate reader (Molecular Devices; Sunnyvale, CA, USA).

### 4.7. Statistical Analysis

The data obtained from the experiments are presented as mean ± standard deviation. The data were analyzed using Microsoft Excel for Mac ver. 16.78 (Microsoft Corp., Redmond, WA, USA); one-way analysis of variance followed by Tukey’s post hoc test were performed in SPSS version 20 (SPSS, Chicago, IL, USA). Differences with *p*-values * < 0.05 and ** < 0.01 were considered significant.

## 5. Conclusions

This study demonstrates that long-term blue light exposure promotes weight gain in mice through increased fat accumulation. Mechanistically, blue light suppresses the expression of Bmal1/Clock, thereby controlling Sirt1/mTORC1/AREBP1 signaling and activating fat synthesis. TA was also shown to improve the expression of Bmal1/Clock (Figure 5). However, the effect of TA on clock genes is unknown, and further research is needed. Furthermore, because blue light and sleep are closely related, it is necessary to examine weight gain caused by sleep and the potential modulatory effect of TA. As this study was conducted in mice, the translational relevance to humans remains uncertain and requires clinical validation. Given the ubiquitous exposure to blue light from screens and devices in modern society, this study underscores the importance of understanding and mitigating the health risks associated with chronic blue light exposure.

## Figures and Tables

**Figure 1 ijms-26-05554-f001:**
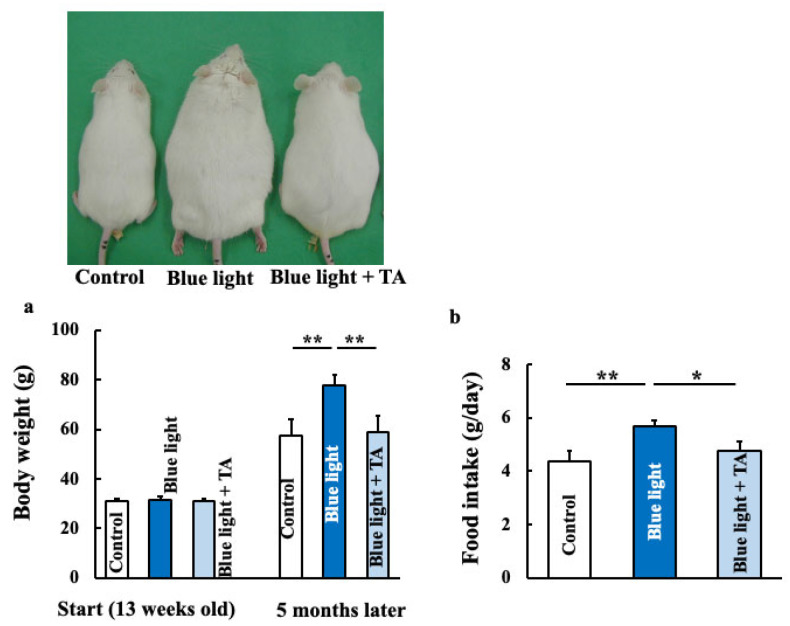
Effects of long-term blue light exposure on mouse body weight (**a**), food intake (**b**), fat accumulation (**c**–**e**), blood triglycerides (**f**), and cholesterol (**g**). Fat accumulation was measured using a 3D micro X-ray CT system, with body fat, subcutaneous fat, and visceral fat being measured separately. Subcutaneous fat is shown in orange and visceral fat is shown in yellow. Triglycerides and cholesterol were measured using commercially available kits. The values are expressed as means ± SD derived from five specimens. * *p* < 0.05; ** *p* < 0.01. TA: tranexamic acid.

**Figure 2 ijms-26-05554-f002:**
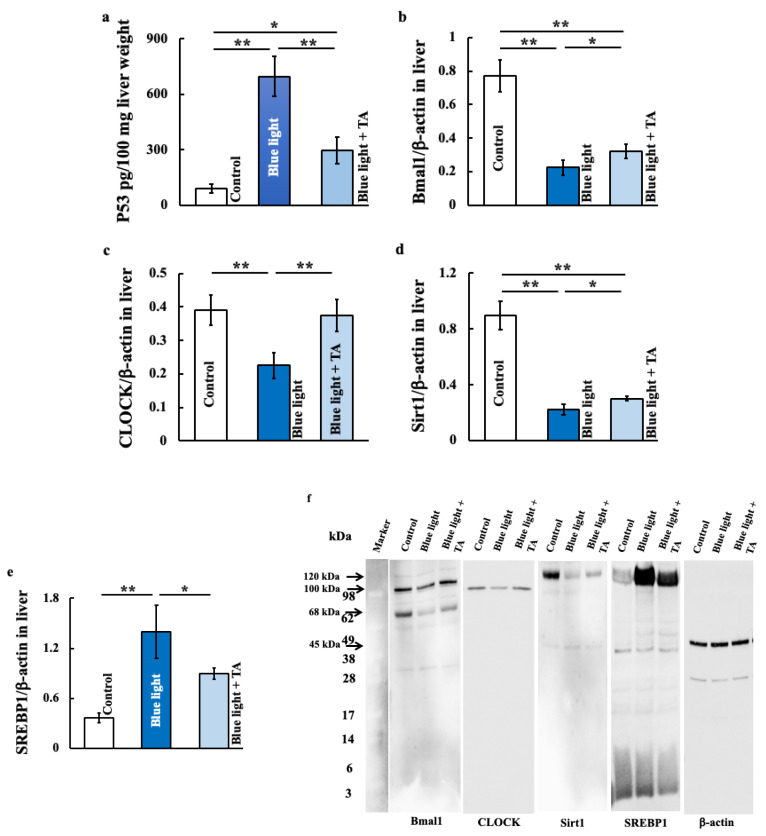
Effect of long-term blue light irradiation on the expression of p53 (**a**), Bmal1 (**b**), CLOCK (**c**), Sirt1 (**d**), and SREBP1 (**e**) in the liver. The expression of p53 was measured using an ELISA kit, and the expressions of Bmal1, CLOCK, Sirt1, and SREBP1 were measured using Western blotting. Western blot diagram of Bmal1, CLOCK, Sirt1, and SREBP1 with molecular weight markers (**f**). The arrows indicate the molecular weight of each protein based on the markers. The values are expressed as means ± SD derived from five specimens. * *p* < 0.05; ** *p* < 0.01. TA: tranexamic acid.

**Figure 3 ijms-26-05554-f003:**
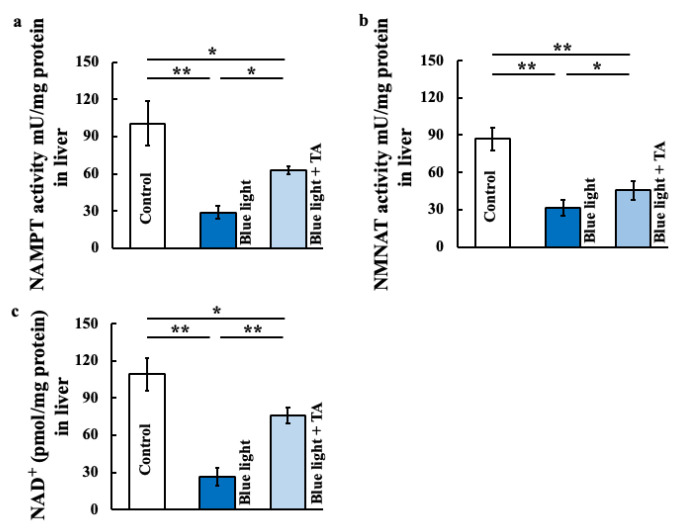
Effect of long-term blue light irradiation on the level of NAMPT (**a**), NMNAT (**b**), and NAD^+^ (**c**) in the liver. The levels of NAMPT, NMNAT, and NAD^+^ were determined using an ELIZA kit. The values are expressed as means ± SD derived from five specimens. ** p* < 0.05; ** *p* < 0.01. TA: tranexamic acid.

**Figure 4 ijms-26-05554-f004:**
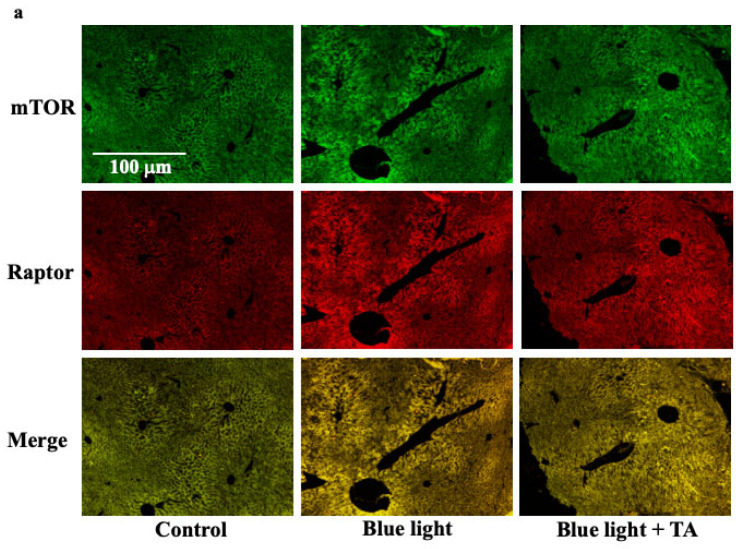
Effect of long-term blue light irradiation on the expression of mTORC1 (**a**), LXRa (**b**), LKB1 (**c**), and AMPK (**d**) in the liver of the mouse specimens. Because mTORC1 is a combination of mTOR and Raptor, we measured the area where mTOR and Raptor merge (*r* = 0.862). The values are expressed as means ± SD derived from five specimens. * *p* < 0.05; ** *p* < 0.01. Scale bar = 100 μm. TA: tranexamic acid.

**Figure 5 ijms-26-05554-f005:**
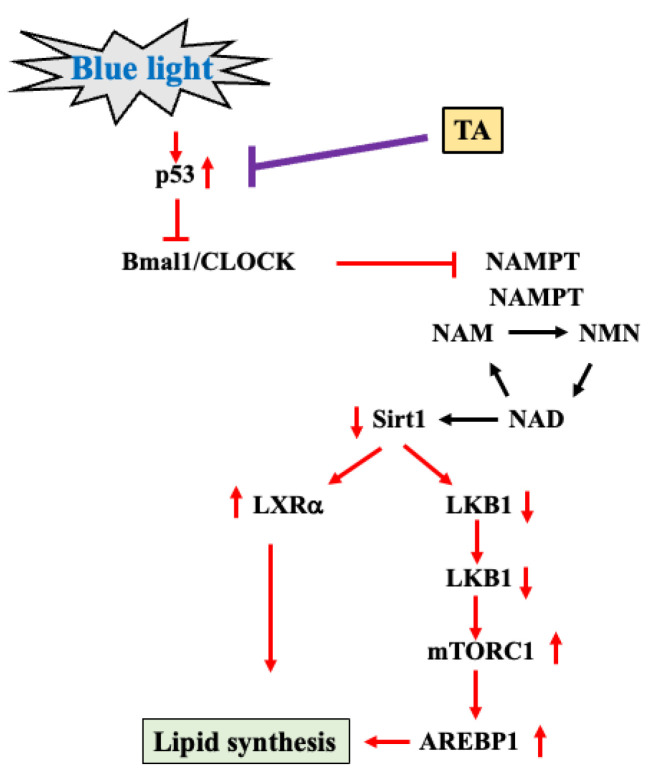
Mechanism of weight gain in mice exposed to long-term blue light and location of TA action. The red arrows indicate signal transduction upon blue light irradiation. The arrows on each protein indicate the increase or decrease in protein level. TA: tranexamic acid.

## Data Availability

Data is contained within the article and Appendix A.

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
