# Peer review of "Effects of Long-Term Blue Light Exposure on Body Fat Synthesis and Body Weight Gain in Mice and the Inhibitory Effect of Tranexamic Acid"

_ijms, 2025, doi:10.3390/ijms26125554_

Round 1
Reviewer 1 Report
Comments and Suggestions for Authors
In this paper, the authors investigated the effect of blue light on the body fat synthesis and weight gain in mice. The manuscript describes the possible pathways affected by the blue light by quantifying different biomarkers, including fat masses, clock-related genes, and liver-related biomarkers. The manuscript also described the inhibitory effect of Tranexamic acid (TA) on blue light-treated mice. Overall, it demonstrated that blue light exposure could cause mice body weight gain by affecting several cellular processes, and TA could efficiently reduce the weight gain caused by blue light. However, there are a few concerns that need to be addressed before recommending for publication.
- In the introduction part, the author specifically mentioned that women would gain more weight under blue light exposure. However, in the experimental design, all mice are male. Is the weight gain caused by blue light related to sex, based on the previous literature? If it is, at least female mice should be added to the experimental plan to investigate the relationship between body weight gain and sex. If not, then the introduction should be carefully revised to avoid confusion and misleading information.
- The introduction needs more proper references cited. For example, the study by Park et al was not cited in the original paper. Please check the introduction part and add more references to support the statements.
- The authors investigated the effect of blue light exposure on liver biomarkers related to body weight gain. Considering the TA was dosed three times a week at 12mg/kg, did the authors investigate the liver damage-related biomarkers? The blue light and the TA treatment may bring more liver damage.
- For Figure 4, the authors mentioned that the intensity measurement of mTORC1 was by measuring the mTOR and Raptor merged regions. What is the correlation coefficient of the regions the author selected for the intensity quantification?
- The discussion of the results simply describes the data collected. More scientific discussion should be included, such as more statistical numbers and what these numbers represent scientifically.
Author Response
To Reviewer 1
Thank you very much for your detailed and useful comments. We have carefully considered the reviewer's point of view. We have addressed each of them as follows.
Comment 1: In the introduction part, the author specifically mentioned that women would gain more weight under blue light exposure. However, in the experimental design, all mice are male. Is the weight gain caused by blue light related to sex, based on the previous literature? If it is, at least female mice should be added to the experimental plan to investigate the relationship between body weight gain and sex. If not, then the introduction should be carefully revised to avoid confusion and misleading information.
Response 1: Thank you for your comment. Males were used because no sex differences were observed in the change in body weight due to blue light exposure. We will add the results to the suppl data.
Line 61-64
We further checked whether there were sex differences in weight gain. As a result, we found that there was no difference between males and females in the effect of blue light exposure on body weight (Figure S1). Therefore, in this study, we decided to use males, considering the effect of the sexual cycle in females.
Figure S1
Comment 2: The introduction needs more proper references cited. For example, the study by Park et al was not cited in the original paper. Please check the introduction part and add more references to support the statements.
Response 2: Thank you for your comment. Park AM on line 32 of Introduction is a mistake; it should be Park YMM. It corresponds to reference 2.
Line 32
Park AM – Park YMM
Comment 3: The authors investigated the effect of blue light exposure on liver biomarkers related to body weight gain. Considering the TA was dosed three times a week at 12mg/kg, did the authors investigate the liver damage-related biomarkers? The blue light and the TA treatment may bring more liver damage.
Response 3: Thank you for your important comments. We measured plasma AST and ALT levels to examine liver damage in mice. As a result, blue light exposure and TA administration had no effect on AST and ALT levels, and did not induce liver damage.
Line 239-244
To examine whether blue light exposure and TA administration would cause liver damage in mice, we measured plasma aspartate transaminase (AST) and alanine transaminase (ALT) levels, which are markers of liver damage. As a result, blue light exposure and TA administration had no effect on AST and ALT levels, and no liver damage was induced (Figure S4).
Comment 4: For Figure 4, the authors mentioned that the intensity measurement of mTORC1 was by measuring the mTOR and Raptor merged regions. What is the correlation coefficient of the regions the author selected for the intensity quantification?
Response 4: Thank you for your comment. The correlation coefficient between mTOR and Raptor is r = 0.862. Added to figure legend 4.
Line 133
Comment 5: The discussion of the results simply describes the data collected. More scientific discussion should be included, such as more statistical numbers and what these numbers represent scientifically.
Response 5: Thank you for your comment. A more detailed discussion of the results is provided in the first paragraph of the discussion.
Line 138-146
Blue light exposure reduces the expression of the clock genes Bmal1 and Clock. Bmal1/Clock suppresses the expression of NAMPT, thereby inhibiting the metabolism of NAM to NMN. In addition, the production of NAD is reduced, thereby inhibiting the production of Sirt1. The reduction in Sirt1 activates LXRa and simultaneously reduces LKB1. The reduction in LKB1 increases mTORC1/AREBP1 and increases lipid synthesis. The increased lipid synthesis is thought to have caused the weight gain of the mice. TA is proposed to suppress the increase in p53, which is mostly located upstream of blue light exposure, and increases Bmal1/Clock, thereby suppressing lipid synthesis.
Reviewer 2 Report
Comments and Suggestions for Authors
The manuscript "Effects of Long-Term Blue Light Exposure on Body Fat Synthesis and Body Weight Gain in Mice and the Inhibitory Effect of Tranexamic Acid" presents an original research in vivo considering effect of blue light on body mass gain.
Comments and questions:
- Introduction: The additional paragraph with information about the tranexamic acid structure, application, physiological properties will be in advance to the presentation;
- The groups treated only with TA were not reported or are studied as controls;
- The investigation looks like communication and need more detail studies in order to proof that the blue light is the reason (at similar food portion);
Author Response
To Reviewer 2
Thank you very much for your detailed and useful comments. We have carefully considered the reviewer's point of view. We have addressed each of them as follows.
Comment 1: Introduction: The additional paragraph with information about the tranexamic acid structure, application, physiological properties will be in advance to the presentation;
Response 1: Thank you for your comment. We have added the structure, uses, and physiological properties of TA.
Line 50-57
In addition, tranexamic acid (trans-4-aminomethylcyclohexanecarboxylic acid) is a medical amino acid with antiplasmin activity, which exerts hemostatic, anti-inflammatory, and anti-allergic effects. The action of tranexamic acid depends on the binding of plasmin and plasminogen to fibrin. As a result, tranexamic acid exhibits fibrinolysis inhibition, arachidonic acid dissociation inhibition, prostaglandin and leukotriene production inhibition, active oxygen dissociation inhibition in neutrophils, and histamine dissociation inhibition in mast cells [14-16]. In addition, tranexamic acid has the effect of reducing age spots, melasma, and pigmentation caused by ultraviolet (UV) rays [17, 18].
- Abiko Y.; Iwamoto M. Plasminogen-Plasmin system. VII. Potentiation of antifibrinolytic action of a synthetic inhibitor, tranexamic acid, by alpha 2-macroglobulin antiplasmin. Biochim. Biophys. Acta 1970; 214, 411–418.
- Chang W.C.; Shi G.Y.; Chow Y.H.; Chang L.C.; Hau J.S.; Lin M.T.; Jen C.J.; Wing L.Y.; Wu H.L. Human plasma induces a receptor-mediated arachidonate release coupled with G protein in endothelial cells. Am. J. Physiol. 1993, 264, C271–281.
- Weide I.; Tippler B.; Syrovets T.; Simmet T. Plasma is a specific stimulus of the 5-lipoxygenase pathway of human peripheral monocytes. Thromb. Heamost. 1996, 76, 561–568.
- Liao L.L.; Pang G.Y. Effect observation of acidum tranexamicum on treating chloasma. Chin. J. Dermatovenereol. 2006, 20, 675–676.
- Li D.; Shi Y.; Li M, Liu J.; Feng X. Tranexamic acid can treat ultraviolet radiation-induced pigmentation in guinea pigs. Eur. J. Dermatol. 2010, 20, 289–292.
Comment 2: The groups treated only with TA were not reported or are studied as controls;
Response 2: Thank you for your comment. The group that received only TA was not included this time because no difference was observed between the group and the control group.
Line 232-235
Furthermore, no change was observed in the control (white light) + TA administration group compared with the control group; therefore, only three groups were included in the study: control, blue light irradiation group, and blue light + TA administration group.
Comment 3: The investigation looks like communication and need more detail studies in order to proof that the blue light is the reason (at similar food portion);
Response 3: Thank you for your comment. In this study, in addition to blue light, we also tested green and red light. However, this phenomenon only occurred under blue light. We also tested bitter foods, just like TA, but the effect was not as pronounced. We have not yet been able to prove that blue light is the cause, and we would like to conduct more detailed research in the future.
Round 2
Reviewer 2 Report
Comments and Suggestions for Authors
The manuscript was improved according to the comments.
Author Response
Comment: The manuscript was improved according to the comments.
Response: Thank you for your comments.
We will use the reviewers' comments to further advance our research.